# Effects of Transient Loss of Vision on Head and Eye Movements during Visual Search in a Virtual Environment

**DOI:** 10.3390/brainsci10110841

**Published:** 2020-11-12

**Authors:** Erwan David, Julia Beitner, Melissa Le-Hoa Võ

**Affiliations:** Scene Grammar Lab, Department of Psychology, Theodor-W.-Adorno-Platz 6, Johann Wolfgang-Goethe-Universität, 60323 Frankfurt, Germany; beitner@psych.uni-frankfurt.de (J.B.); mlvo@psych.uni-frankfurt.de (M.L.-H.V.)

**Keywords:** visual attention, visual field loss, gaze-contingent protocol, visual search, virtual reality

## Abstract

Central and peripheral fields of view extract information of different quality and serve different roles during visual tasks. Past research has studied this dichotomy on-screen in conditions remote from natural situations where the scene would be omnidirectional and the entire field of view could be of use. In this study, we had participants looking for objects in simulated everyday rooms in virtual reality. By implementing a gaze-contingent protocol we masked central or peripheral vision (masks of 6 deg. of radius) during trials. We analyzed the impact of vision loss on visuo-motor variables related to fixation (duration) and saccades (amplitude and relative directions). An important novelty is that we segregated eye, head and the general gaze movements in our analyses. Additionally, we studied these measures after separating trials into two search phases (scanning and verification). Our results generally replicate past on-screen literature and teach about the role of eye and head movements. We showed that the scanning phase is dominated by short fixations and long saccades to explore, and the verification phase by long fixations and short saccades to analyze. One finding indicates that eye movements are strongly driven by visual stimulation, while head movements serve a higher behavioral goal of exploring omnidirectional scenes. Moreover, losing central vision has a smaller impact than reported on-screen, hinting at the importance of peripheral scene processing for visual search with an extended field of view. Our findings provide more information concerning how knowledge gathered on-screen may transfer to more natural conditions, and attest to the experimental usefulness of eye tracking in virtual reality.

## 1. Introduction

In the natural world, we make a succession of fixations by which we process information from the central and peripheral visual fields to explore our environment, usually in pursuit of a behavioral goal. In this study, we aimed at understanding the impact of vision loss during a search task carried out in a virtual environment replicating natural, everyday rooms. Do results from on-screen experiments translate fully to a situation where participants can use more than a screen-worth of visual field, while moving their neck and body along with their eyes? The advent of virtual reality (VR) head-mounted displays with low frame rate latency, acceptable graphic resolution and embedded eye tracking systems, now allows researchers to implement protocols that would not be possible with mobile eye tracking in the real world. We took this opportunity to study how head movements serve visual search and to measure visuo-motor behaviors across search phases.

The central and peripheral fields of view differ by the density of photoreceptor cells [1,2], size of receptive fields [3,4,5], as well as their cortical representation [6,7]. As a result visual information quality drops as a function of the eccentricity to the fovea [8,9]. Despite its limitations, peripheral vision helps build a global representation of scenes [10,11]. The work of Boucart et al. [11,12,13] served to demonstrate that scene categorization stays above chance-level even at high visual eccentricities (70 deg. and farther). Functionally, peripheral vision serves to explore a scene, while central vision is used to sequentially analyze regions of interest. This manifests itself in different types of eye movement control, e.g., local and global processing [14,15,16], where global processing is exploratory and characterized by long saccades and short fixation durations, whereas local processing is reflected through short saccades and longer fixation durations required to analyze and identify objects or regions of interest. Object features, such as color, help localize search targets in natural scene peripherally [17], whereas, central vision exploits color to identify targets; contrary to textbook knowledge, color perception is indeed effective for scene processing at mid to high eccentricities (approximately 50 deg.) [18,19].

The dual role of the visual field is further demonstrated by the visuo-motor capabilities of participants experiencing transient (e.g., [20,21,22,23,24]) and pathological [25,26] central field defects. The complex consequences of real vision losses notwithstanding, lack of central vision results in diminished performances compared with normal sighted observers. Similarly, transient peripheral vision loss strongly impacts eye movements and search performances [20,21,22,23,24]. The current understanding is that the entire visual field is necessary to carry out visual tasks, but central and peripheral vision most likely have different roles when interacting with naturalistic environments.

Real-world scenes and everyday goals deviate from laboratory experiments in many aspects, for instance with regard to the complexity of the situations [27]. So far there is evidence that overall, visual search findings gathered with computer set-ups translate well to real-world situations [28,29,30,31]. In this study, we constructed VR environments by replicating real-world indoor scenes. Combined with the ability to freely move around thanks to the head-mounted display, we intended to approach quasi-natural conditions. In VR and mobile eye tracking experiments, neck and body movements need not be restricted; as a result, observers use the full range of movement permitted by the human body to explore their surroundings.

Due to the addition of head and body movements, tracking eye movements in VR environments reveals strong differences compared to eye movements recorded in restricted on-screen paradigms. The field of fixation is the area within the field of view where fixations can be made only with the eyes, when the head is restricted [32,33,34]. In cases where the head is unrestricted this becomes the practical field of fixation [34,35]. We do not make use of the entire field of fixation in normal situations, thanks to head movements permitting large saccades without resorting to long eye rotations [36,37,38,39,40]. Head movements are synergistic when they contribute to saccades [39] and compensatory when they help stabilize gaze in the scene during fixations (vestibulo-ocular [39,41] and opto-kynetic [42] (Chapter 2) [43] responses). During synergistic movements, head movements are recorded even during small saccades [39,44,45,46]. In a free-viewing study in VR [47], we observed that head movements were much less complex compared to eye movements; possibly under the control of behavioral goals related to exploration, whereas eye movements served to analyze regions of interest.

This complex interplay of head, body, and eye movements allows for efficient processing of the visual world that surrounds us. So what role does central vs. peripheral processing play during unrestricted, real-world search of 3D environments? For this purpose, we employed a gaze-contingent protocol to remove part of the field of view in real-time. This protocol [48,49] has been extensively used to segregate central and peripheral visions. The principle idea is to present contents of different natures in the different fields of view. A moving mask (central masking) or moving window (peripheral masking) is centered on the gaze position sampled by an eye tracking device. In the 1970’s, it was introduced to study reading [50,51]. It has since been used in conjunction with natural scene viewing tasks to study eye movements, visual perception and visual attention (free-viewing [24,47], scene exploration [20,23,52] and visual search [17,21,53,54]).

While these studies have been essential in furthering our understanding of central vs. peripheral processing, they have done so in very restricted, 2D scenarios. Employing modern VR devices, however, allows researchers to study visual attention in quasi-natural situations. Advancing to VR resolves some shortcomings of gaze-contingent protocols in 2D: the portion of the field of view excited is significantly increased, masking can be applied independently per eye, and neck and body movements are unrestricted. An eye tracker system embedded in a VR headset allows the implementation of protocols that would not be possible with mobile eye tracking devices, for instance, manipulating the visual fields during natural viewing. Virtual reality devices have significant advantages over mobile eye-tracking. Apart from the possibility to fully manipulate the content of the field of view they also offer the opportunity to generate many complex scenes and easily manipulate them according to various experimental conditions. Additional, contrary to most mobile eye-tracking studies we here report gaze data as the combination of head and eye movements, instead of eye movements alone which would withhold important information, such as synergistic and compensatory head movements (e.g., [39]). Some VR devices possess two cameras on the front that can project a stereoscopic view of the real world inside the headset, allowing a gaze-contingent modification of a projection of a real environment in real-time. On the other hand, gaze-contingent protocols in VR come with new disadvantages compared to on-screen studies: the eye-movement-to-display-update latency increases slightly, stimuli resolution is lowered, eye tracking accuracy can be reduced. Overcoming these obstacles in order to investigate the roles of central vs. peripheral processing during the search of complex 3D environments was the aim of the present study.

### The Present Research

In this study, we measured the effect of transient vision losses on visuo-motor behaviors in VR as a mean of studying the role of central and peripheral vision for visual search in more natural settings. We compared our results to previous on-screen experiments to determine how much of what is learned on-screen can be transferred to natural viewing. For that purpose, we implemented a gaze-contingent protocol to display in real-time central and peripheral masks. One novelty in particular, was the use of a head-mounted display retrofitted with a binocular eye tracker to display transient loss of vision while exciting a large part of the field of view. We analyzed the role of central and peripheral visions in a VR environment where users were able to freely move while searching for objects. In a VR context, we report eye (eye-in-head), head (headset tracking in space) and gaze (eye-in-space) movements. The latter are the final combined gaze movements in the 3D environment. In this paper, we focused mainly on reporting the effects of masking on visuo-motor measures during search in 3D environments.

In a first step, we analyzed the impact of central and peripheral masking on visuo-motor behaviors (hypotheses *Ha*). On-screen literature has extensively demonstrated that saccades target areas where information is best [52]. In effect, average saccade amplitudes increase when central information is modified whereas they decrease when peripheral information is modified [8,20,21,22,23,24,47,53,54]. In a VR experiment, where the field of view is extended and the head is free to rotate, the same effects were reported for eye and gaze movements while free-viewing [47]. Based on these findings, we expected average eye and gaze amplitudes during saccades to display the same effects also during visual search in VR. As reported in free-viewing, head movements decreased with a mask [47], maybe in an effort to control an unusual environment; we expected head movements to be similarly impacted in our analyses. Fixation durations increased significantly when part of the visual field was altered, reflecting cognitive processing difficulties when analyzing modified central information and planning new saccades when peripheral information was modified [20]. In pure free-viewing tasks, fixation durations have been shown to decrease when the central field of view was masked due to an increase in return saccades [24,47]. On the other hand, large peripheral masks reduced fixation durations, whereas smaller masks resulted in an increase. Besides the difference in task, the nature of the masking plays an important role. Cajar et al. [23] implemented low and high-pass filtering masks and argued that when processing becomes too difficult, such that information cannot be efficiently used for gaze control, the duration of fixations does not further increase. Average fixation durations increased when central information was deemed useful (high spatial frequencies in their case). Since we implemented masks removing all visual information, we expected fixation durations to decrease in the absence of central information to process (central masking) and increase without peripheral stimulation to reflect difficulties in saccade planning. Finally, gaze-contingent masks affect saccade directions as well. Namely, central masking increases return saccade rates while peripheral masking diminishes them [24,47,55,56]. This is observed in free-viewing [24,47] and visual search [56], on-screen [24,56] and in VR [47], therefore it appears to be driven by visual stimulation rather than a top-down effect. Moreover, David et al. [47] showed that this effect is exhibited by eye movements rather than head movements. As such, we expected to observe a similar increase of return saccade rates with central masking and a decrease with peripheral masking through eye and gaze movements, as well as a strong tendency toward forward relative movements by the head.

Free viewing and visual search are often used to study visual attention. Visual search is a highly dynamic task, which can be subdivided into three temporal phases [57]. The initiation phase is the duration of the first fixation at trial onset, and seems to represent the period during which scene’s gist is being processed. During the subsequent scanning phase, participants explore the scene in search of a predefined target object. This phase ends with the fixation preceding the first saccade made on the target object. The fixation of the target initiates the verification phase. This last phase accounts for the identification of a target, usually done by foveating it and lasts until the participant presses a button to formally select an object as the expected target. Hypotheses *Hb* encompasses the measurements of average saccade amplitude and fixation duration as a function of these last two phases. With regard to main effect of search phases we hypothesize that the scanning phase is driven by exploration and therefore will be mainly represented by short fixation and long saccades, while the verification phase reflects processing of fine information on the fovea and requires longer fixations and shorter saccades. We expected the interaction of masking and search phase to result in a stronger increase in backward saccades when central vision is missing during verification phases; because in this phase, observers need central vision to make a decision, a central mask will make this harder. The fact that this phase relies heavily on fine perception via central vision should logically result in an increase in return saccade rates as participants attempt to perceives objects foveally in spite of the mask. Considering phases of visual search is important to understand the dynamics of this task. To the best of our knowledge, this is the first study identifying exploration and analysis behaviors in relation to search phases. The opportunity to record head movements lets us measure how the eyes and the head contribute independently to these behaviors.

The study plan and analyses were preregistered on the Open Science Framework website (https://osf.io/g9npw/?view_only=53eca7be33684b0cad4d220f59b90615). The second set of hypotheses related to search phases was not initially intended as per the preregistration document. As such, statistical analyses associated to them should be considered unplanned.

## 2. Method

### 2.1. Participants

Thirty-two fluent German speakers were recruited for this experiment (24 women, mean age: 24 years old, minimum: 18, maximum: 43). Recruiting was stopped earlier than planned in the pre-registration document (the aim was to test 45) due to COVID-19. Participants with eyeglasses were not included in this study because frames and lenses reduce the eye tracker’s accuracies. We asserted normal visual perception by testing acuity (Landolt “c” chart), color perception (Ishihara color blindness test) and depth perception (stereo fly test). In order to improve comfort and viewing conditions we measured interpupillary distance of participants and set the measurement as the distance between the VR headset’s lenses [58]. Finally, we asserted the dominant eye with the cardboard or thumb test. All participants gave their written consent before beginning the experiment and were compensated for their time with university credit or eight Euro/h. The experiment took a maximum of 60 min, conformed to the Declaration of Helsinki, and was approved by the local ethics committee of the Faculty of Psychology and Sport Sciences at Goethe University Frankfurt.

One participant used a VR headset occasionally, one had used a headset twice before, three had once before, the rest of the participants had never used a VR headset before the experiment. One participant reported motion sickness during the experiment, but chose to finish after resting.

### 2.2. Apparatus

Virtual scenes were displayed inside an HTC Vive (HTC, Valve corporation) VR head-mounted display worn by participants. The display is refreshed 90 times per second and shows, on average, a field-of-view of 90 by 90 deg. binocularly. The headset was retro-fitted with a Tobii eye tracker (Tobii Technology), tracking gaze binocularly at 120 Hz with a precision below 1.1 deg. within a 20 deg. window centered in the viewports.

The experimental protocol was implemented in the Unity game engine (Unity Technologies) with the Steam VR (Valve corporation) and Tobii eye tracking (Tobii Technology) software libraries. We minimized the time between gaze sampling to updating a mask position in a viewport, by updating the display as late as possible in the rendering process with the latest sample received from the eye tracker. The latency problem is critical in moving-mask conditions, because we need to ensure that central vision is modified even at the start and end of saccades [49,59]. We estimated the maximum (“worst-case scenario”) latency to be below 15 ms.

### 2.3. Stimuli

Sixteen natural everyday rooms were used as virtual scenes (Figure 1). The rooms included three living rooms, bedrooms, bathrooms, and kitchens as well as four offices. Their size was chosen to fit in the physical room where the experiment took place (3.8 by 3.5 m), so that there was no risk for participants of colliding with walls or furniture. The room and object dimensions were chosen to be similar to the real-world. Virtual scenes were populated with 36 unique objects keeping in mind scene grammar rules. Scenes were globally illuminated. As a result, objects did not cast shadows and looked similar from any angle. For a complete description of the scenes please refer to [31]. One of the scenes was set aside for a training phase.

A lexicon of German words was built, referencing all objects present in the scenes. Words from this lexicon were used when displaying a target word relating to the object to look for in the scenes. We chose to use target words rather than pictures of the target in order to reduce search guidance based on low-level, visual characteristics of the actual target, in favor of more high-level guidance by scene grammar. For each scene, six target objects were selected.

### 2.4. Experimental Design

We implemented a gaze-contingent protocol in order to remove central or peripheral visual information where a participant’s gaze was located in real-time (Figure 2). In a shader pass, we masked part of the scene by adding a gray circular mask centered on gaze positions to obtain central masking. The inverse situation (only a central circular area remained unmodified) was implemented to remove information in the peripheral field of view. Alpha-blending was used at the edge of the circle to create a smooth transition between the mask and the scene [60]. This method was applied to both eyes: the left viewport was updated with left gaze data and the right one with right gaze data.

We chose a circle of six deg. of radius for both central and peripheral masking conditions on the basis of results from a previous experiment [24], and a pre-study dedicated to testing the implementation of the protocol (six subjects, 540 trials). A smaller radius was not advised, as we started noticing latency effects [49,59]. Moreover, a bigger radius may provoke results indistinguishable from a no-mask condition as it approaches the visual span required to complete visual search tasks (eight deg. [54], though this figure was obtained on-screen). We selected only one radius in order to keep sufficient statistical power. The experiment therefore contained two masking conditions (central and peripheral masks) as well as a no-mask condition (control).

Participants were tasked with finding six target objects in 16 scenes, one per trial in one of the three mask conditions (90 trials plus six training trials). We controlled for semantic congruity when selecting target objects. The announced target object was always present in the scenes Figure A1 (Appendix B) shows the polar distribution of target directions relative to the observer’s starting position and direction in scenes. We counterbalanced our experimental design so that target objects, scenes and mask conditions across all subjects would be presented the same number of times in a different order.

### 2.5. Procedure

The experiment started with a calibration procedure (nine points), followed by a validation procedure (nine points) in order to calibrate and test eye tracking precision within 20 deg. from the center of the viewports. We did not test past this window because it has been shown that observers within VR headsets very seldom look further in the periphery of the viewports [47,61,62]. Mean successful validation accuracies across subjects was high (95% Confidence Interval (CI) [1.67, 2.21]). More information about the calibration/validation phase can be found in the Appendix A. Participants got accustomed to the material and the protocol in a training phase where they searched for six objects in one dedicated scene that was not used in the rest of the experiment. They experimented with masked and control conditions (twice each in a random order).

A trial went as follows: starting in an empty room, a black screen on a wall displayed instructions. Participants had to walk onto a blue square on the floor, this was the starting search position from which all target objects could be seen and where a user’s body did not intersect with any object (close to the center of the rooms). Then, they had to fixate the screen for one second before the target word for that trial appeared (in German). The word was displayed for a second before it disappeared along with the pre-trial room, instantly replaced by the virtual scene. Observers had 30 s to find a target object. They could end a trial early by pressing the trigger on the VR controller (held in their hand, but invisible in the virtual world). They were told not to aim at the target object with the controller but to look at it. That allowed us to register from their gaze if they had identified the right object or not, while reducing a bias related to the time needed to target an object in the virtual world. Trials were separated by at least one second (as the program was loading the next scene), a calibration/validation phase was triggered every 10 trials, a resting period was inserted every 30 trials but participants could take a break after any trial. Participants went through all trials in 40 min on average.

### 2.6. Data Preparation

During trials we retrieved the following data: head rotation (quaternions), left and right gaze directions (3D unit vectors), left and right eye positions (2D position in the left and right viewports), and the object hit by left and right gaze rays (lines in 3D space starting from the left or right eye position, going in the direction of the gaze vector). In order to process eye, head and gaze data we adapted the toolbox [63] developed for the Salient360! benchmark [64,65].

Gaze unit vectors are expressed relative to camera rotations. Therefore, we start by multiplying camera rotations with gaze vectors (quaternion-vector multiplication) to obtain combined eye and head gaze directions in 3D space. This representation and head rotations are projected on a unit sphere centered on the user’s head position. Eye movements are 2D positions on the viewports (views inside the headset projected on the displays). We will refer to the addition of head and eye rotations as the **gaze** data (*eye-in-space*), whereas the position of the gaze (dominant eye) in the viewport (*eye-in-head*) will be **eye** data, head rotations are **head** data [66,67].

We identified saccades and fixations on the basis of the combined gaze signal with the help of a velocity-based algorithm [68]: we obtained velocity between gaze samples (deg. per ms.) by calculating the orthodromic distance (great-circle distance) and dividing by the time difference; the resulting signal was smoothed with a Savitzky-Golay filter [69]; filtered samples below 120 deg./ms. were identified as part of a fixation. Visual search phases are identified on the basis of the combined gaze data.

We removed trials for which more than 15% of either left or right eye tracking signals were lost (N = 77). In the context of our protocol, loss of signal results in perceiving part of the scene when that should not be possible. We removed trials that lasted less than 500 ms. or for which only one fixation was found, as they were probably the result of a misfire (N = 2). From this dataset we extracted eye, head and gaze movement-related variables which we describe and analyze in the next section.

## 3. Analyses

Continuous response variables were analyzed with linear mixed models [70] in order to account for fixed and random experimental effects. The random effects present in our experiment are subjects, scenes and objects (present as random intercepts). We tested *Ha* and *Hb* hypotheses with planned contrasts:*Ha*: control condition vs. central mask vs. peripheral mask;*Hb*: masking conditions as a function of search phases (scanning, verification).

Hypotheses *Hb* were also tested descriptively by plotting measures over time (Appendix E), and we also displayed measures over a normalized time-course expecting regularities to appear (Appendix D). We believe that a short scanning or verification period does not necessarily compare to a longer one or a failed trial (when a target was not found in the allotted time). As a result, we separated scanning and verification periods into phases lasting less than 2.5 s, less than 7 s, over 7 s, and failed trials (Figure A3, Appendix D).

As part of our analyses we report the following values: *b*-value (estimated difference between means), SE (estimated standard error) and *t*-value. An asterisk next to a variable’s name in Figure 3 means that distributions were log-transformed to improve the normality of the residuals and the reliability of the models. Analyses were run in *R* (The *R* Project for Statistical Computing [71]) using the *lme4* package [72]. Statistical information about the second set of hypotheses is reported in Table 1.

We removed fixations lasting less than 80 ms. (N = 308) as we consider this the minimum amount of time needed to analyze visual information and prepare the next saccade [42,73,74]). Fixations lasting more than 2000 ms were also removed (N = 20) because they have a high probability of being caused by processes independent from the experimental task [75]. The final dataset is made up of 37,111 fixations and 36,950 saccades.

### 3.1. Fixation Durations

Fixation durations are affected by the exploitation of central information, as well as by the exploration of the peripheral field of view needed to prepare the next saccade [17,23,76,77]. Past screen-based studies have shown that degrading or removing information within the field of view resulted in longer fixation times as scene processing is made more difficult and more time is needed to analyze central or peripheral data [17,20,23]. This effect is particularly salient when peripheral vision is modified.

#### 3.1.1. Central vs. Peripheral Masking

When central vision was impaired fixations lasted on average approximately as long as in the no-mask condition (188 ms; b=0.01,SE=0.01,t=1.48; Figure 3g). The absence of peripheral stimulation reduced fixation durations to 177 ms (b=−0.04,SE=0.01,t=−7.03). Differences between the scanning and verification phases strongly show an increase in fixation durations reflecting a need for more foveal processing when identifying a target before ending a trial. Figure A6 (Appendix D) shows that over normalized trial time-courses verification trials end with longer fixation durations which are necessary to analyze finely an object of interest and identify it as a target.

#### 3.1.2. Scanning vs. Verification Phase

During scanning, masking had little effect on fixation duration (Figure 4g). However, we observe a decrease in average durations with a mask compared to a no-mask condition during the verification phase. We would have expected the combined effect of masking and verifying to increase fixation durations above the control baseline. In the case of central masking this can be explained by the fact that an increase in return saccades was reported to decrease fixation durations [24,47] because once a region of interest is fixated it is masked and peripheral regions become much more salient in comparison [55], resulting in a fast back-and-forth behavior between a region of interest and the rest of the scene. As for the decrease during peripheral-masking trial, we believe that the mask is large enough not to hinder the task significantly but still shorten fixation durations due to the absence of peripheral information processing.

As expected, scanning is determined by exploration with small fixation durations, while they increase significantly during verification where potential targets are analyzed. We could have expected to observe longer durations during scanning as well because of difficulties in identifying targets in the periphery with a central mask and the absence of periphery information to direct attention in the second masking condition. But our results indicate that target pre-identification in the visual periphery is truly achieved without central vision. As for peripheral masking, the absence of peripheral regions of interest resulted in a strong saccadic momentum, a mechanism of exploration [78,79] whereby saccades are directed roughly in the same direction, thus away from previous fixations.

#### 3.1.3. Visual Search vs. Free-Viewing and On-Screen vs. Virtual Reality

In free-viewing conditions on-screen a decrease in fixation durations was observed when central vision was removed [24]. This effect increased as mask size grew, probably reflecting the increase in return saccade rate observed jointly: with no central information to analyze and little top-down drive brought by the task, participants quickly produced saccades outside of the masked area. Conversely with peripheral masking, masks left more central information to process as the peripheral masks became smaller, leading to increased fixation durations because saccade planning was made difficult by a lack of peripheral information to target. During a free-viewing task in VR [47] durations decreased with central masks of six and eight deg., whereas the impact of peripheral masks of six deg. was similar to a no-mask condition. Four deg. peripheral masks resulted in an increase of fixation durations, possibly showing that fixation duration is inversely correlated to peripheral mask size. These results established that for a wide field of view these large masks produce similar effects compared to very small masks on-screen (1.5 and 2.5 deg. of radius), probably because object sizes are smaller in natural scenes presented on-screen rather than in VR. It appears that a peripheral mask of six deg. in our study replicates the effect of large similar masks on-screen (>3.5 deg.). In addition, compared to on-screen gaze-contingent experiments (e.g., [17,20,22]) fixation durations reported in VR previously [47] and in this study seem shorter (approximately [260–300] ms on-screen and [180–200] ms in VR). This may be the result of a more natural environment, but let us remember that fixation durations are impacted by the choice of gaze-parsing algorithm and its parameters [80,81,82], therefore this result may be an experimental bias and only the within-study differences between conditions should be considered.

### 3.2. Saccade Amplitudes

Saccade amplitudes are reported to be strongly impacted by the removal of parts of the visual field. As observers plan fixations where information is best at the time of saccade planning [23,24,52,83,84,85], peripheral masking results in average amplitudes below mask radius and central masking in average amplitude exceeding mask radius. Interestingly, with a peripheral mask in on-screen experiments, observers targeted the edges of the mask and only went significantly beyond in cases where the mask radius was very small (e.g., 1.5 deg. [24]). Therefore, despite building a representation of the scene, participants do not use that knowledge to plan saccades into the masked periphery. Access to eye movement data collected in 3D VR environments now allows asking whether (1) this effect replicates when the head is unrestricted and (2) how eye and head rotations are impacted, if at all? We measured saccade amplitude as the orthodromic distance between two fixation centroids for gaze and head, and as the Euclidean distance on the viewport for eye movements.

#### 3.2.1. Central vs. Peripheral Masking

Contrary to our expectations, removing central vision resulted only in a small increase in the average eye movements during saccades (b=0.13,SE=0.01,t=16.34; Figure 3a), while we did not observe an effect in general gaze movements (b=0,SE=0.01,t=0.36; Figure 3c). This is the result of a decrease in head movement amplitude (b=−0.11,SE=0.01,t=−10.93; Figure 3b), an effect that was observed in VR previously [47]. This may be explained by the fact that eye and gaze amplitudes during saccades were higher than observed in on-screen experiments (≈4–6 deg.) and in VR (≈6 deg.) to begin with. When peripheral vision was removed, movements were globally reduced: seen via eye (b=−0.88,SE=0.01,t=−119.82), head (b=−0.18,SE=0.01,t=−20.2) and gaze (b=−0.46,SE=0.01,t=−59.29) movements. Once again, gaze-contingent masking lead the eyes to target areas within or at the mask’s boundary (Figure 5b). In effect, observers very seldom executed saccades to locations where there was no information, showing a dependency on visual stimulation. In the case of central masking, making longer saccades is unusual, therefore participants produced the shortest saccades they could outside of the mask. However, the no-mask condition showed that they would not have produced particularly smaller saccades by default; as for peripheral masking, targeting the boundary amounts to maximizing information sampling under the constraint of not reaching outside of the mask.

#### 3.2.2. Scanning vs. Verification Phase

Across scanning and verification time-courses, we notice that at the beginning of scanning phases head rotations are shorter but increase over the course of the first few seconds (Figure 4b, Figure 6, and Figure A8, Appendix E). Moreover, head movement amplitudes decreased further towards the end of the verification phase as the field of view is stabilized to process a potential target. As for eye rotations, they seem to start scanning phases with higher amplitudes and decrease in the first seconds as the head picks up amplitude (Figure 4a). This dynamic appears to originate from head movements; it seems that participants are in a phase of exploration with long head movements before bringing a target into view. What ensues is a sharp decrease in head movement and increase in fixation durations as participants transition from processing the scene globally to processing locally.

During the scanning phase observers made longer head and gaze movements (Figure 4c) than in the verification phase in order to scan the omnidirectional scene in search of the target. This strong effect does not seem to interact with masking conditions. Interestingly, the average eye movement amplitude did not differ in the two search phases. We found no records of similar phase analyses of eye movements in on-screen studies, therefore we cannot compare our results to past findings. Nevertheless, on-screen we would of course also expect average saccade amplitudes to be longer during scanning compared to verification. In the present experiment large shifts of attention were made more efficiently with the head considering the omnidirectional nature of the stimuli, whereas on-screen what would be a small movement in VR (<15 deg.) could traverse half the scene. In essence, the nature of the VR content may be what elicits exploratory head movements. If we presented non-omnidirectional stimuli in VR with the same relative size as on-screen studies, possibly that eyes movements would show exploratory behaviors and head movements would nearly disappear.

#### 3.2.3. Visual Search vs. Free-Viewing

Eye and head movements measured in this experiment were longer than reported in VR for a free-viewing task [47] across all masking conditions (approximately by one deg. for eye amplitude and four deg. for head amplitude). We hypothesize that this reflects a top-down strategy brought out by the search task to quickly find the target by exploring with longer saccades.

#### 3.2.4. On-Screen vs. Virtual Reality

Overall, we replicated the effects of a central versus peripheral gaze-contingent masks on eye movements, however, the ability to use the head in VR to explore resulted in final gaze movements that differed from what we observed on-screen. As can be seen in the saccade direction polar plots (Figure 5), the eyes very rarely escaped the visible areas during peripheral-masking trials, suggesting a strong effect of vision loss on eye movement preparation. Furthermore, it is worth noting that the addition of eye and head movements placed most saccade landing positions just outside of the mask (at the time of saccade planning). If replicated, this would stand in contradiction with theories of trans-saccadic memory requiring an object to target in the periphery [86,87]. On another hand, it seems that deviating from saccadic landing points does not necessarily impact trans-saccadic integration, because attention is not necessarily located at the saccadic landing point [88,89,90]. This observation raises questions in relation to head and eye saccadic planning and trans-saccadic integration: were participants expecting to saccade within the visible area and felt distraught by mismatched expectations, or were head movements taken into account to purposefully target locations partly void of information at the time of planning? The resulting gaze amplitude distribution (Figure 5c) shows most saccades targeting areas just outside of the mask, which means that after a saccade the new unmasked field of view shares half of its content or more with the preceding one, only displaced. In that regard, there may be enough information shared between fixation locations to permit trans-saccadic integration. This could show an optimal sampling solution to targeting as far as possible into a masked region without incurring negative effects of failed trans-saccadic integration; under the condition that eye movement programming be stimulus-driven but head movement can be modulated by the task (exploration behavior).

### 3.3. Relative Saccade Directions

“Relative” here describes the angle between two consecutive saccade directions. In contrast to absolute angle informing about the direction of a saccade compared to the horizontal plane (not analyzed but presented in Figure A2, Appendix C), relative angles inform about the dynamics between saccade directions. Relative saccadic angle distributions are presented as polar plots in Figure 5. In this figure the saccadic momentum [78,79] is identified as a mode at 0 deg. of rotation (East; in the same direction as the previous saccade), whereas the facilitation of return [78] appears as a mode positioned at 180 deg. (West; in the inverse direction compared to a previous saccade direction). Past on-screen [24,55] and VR [47] studies have shown that central masking increases return (backward) saccades because observers make back-and-forth motions between regions of interest that they could not properly analyze because of the mask. By contrast, peripheral masking results in fewer backward saccades as the absence of information in the periphery prevents visual attention from lingering on previously fixated regions, decreasing facilitation of return tendencies to the benefit of saccadic momentum.

We obtain angles between saccade direction by projecting gaze position (on a sphere) on to a plane with a Mercator projection. Contrary to an equirectangular projection, the Mercator one is conformal (it preserves angles). This variable is transformed into a ratio in order to study it statistically with linear mixed models. We consider the saccadic reversal rate (SRR [91]): the number of saccades directed within a 170–190 deg. interval divided by the total number of saccades in a given trial. This ratio measures backward (possibly return) saccades, it represents how often a participant looked back precisely in the direction where a fixation was made at t−1.

#### 3.3.1. Central vs. Peripheral Masking

In accordance with past research, we observed an increased number of return saccades when central vision was masked (eye: b=0.02,SE=0,t=4.05; gaze: b=0.04,SE=0.01,t=8.5; Figure 3d–f). This increase was marginal in the case of head movements (b=0.01,SE=0,t=2.08). Participants persevered in trying to analyze foveally a region of interest: a back-and-forth takes place where a fixation is made on a region of interest which is then devoid of information, making peripheral portions of the field of view much more salient, thus a new fixation is made outside unmasking the previous regions. In contrast, when no peripheral information is available, gaze appeared to be directed predominantly by saccadic momentum: the SRR strongly decreased globally in eye (b=−0.03,SE=0,t=−7.12), head (b=−0.03,SE=0,t=−6.54) and gaze (b=−0.05,SE=0.01,t=−10.38) movement directions. However, we observed that eye relative directions with a peripheral mask were almost uniform radially (Figure 5b). The strong bias in forward saccade rates that can be seen in this condition was driven by the head (Figure 5a).

#### 3.3.2. Scanning vs. Verification Phase

Our analyses of search phases (Figure 4d–f) shows that return saccade rates increased particularly during verification phases when central vision was masked. Return saccade rates were at no-mask level during scanning, showing exploring on par between central-masking and control trials, but shot up in the final phase that required a fine perception of objects for analysis. This increase was apparent in eye rather than head movements. Head movements show a general decrease in the second phase reflecting the stability necessary to finely analyze with the eyes. The absence of peripheral stimulation resulted in fewer return saccades, approximately of the same order in both search phases.

#### 3.3.3. Visual Search vs. Free-Viewing

Compared to a previous study implementing free-viewing in VR [47] our general results (aggregated over search phases) overall do not differ. Scanning and verification represented as periods of exploration and analysis, also manifest themselves in free-viewing [14,92]. While it seems that the general behavior is similar between tasks, more analyses are needed to study global and local scene processing in free-viewing where there is possibly less pressure to explore and/or analyze.

#### 3.3.4. On-Screen vs. Virtual Reality

To our knowledge only one on-screen study, which implemented masking during a free-viewing task, reported a measure of return saccade rates [24]. Therefore, data for a comparison is limited. We can say that the average difference between control and central-masking trials is comparable to on-screen masks of radius 2.5 to 3.5 deg. Whereas the impact of peripheral-masking seems to correspond to a larger mask radius of 4.5 deg.

## 4. Discussion

With a recent push to study vision under increasingly realistic conditions, it has become crucial to take a closer look at eye, head, and gaze movement characteristics during a real-world task in 3D space. Moreover, there is an increasing need to test which findings taken from on-screen studies actually hold truth when moving into 3D and vice versa. In the study presented here, we therefore implemented a gaze-contingent protocol to remove either central or peripheral vision while observers accomplished a real-world search task immersed in virtual every-day scenes. A particularly original aspect of this study is the measurement and comparison of visuo-motor behaviors across visual search phases (scanning and verification). Our results have important implications regarding the reproducibility of knowledge gathered in on-screen experiments for more natural environments. Additionally, to our knowledge this is the first study reporting on visual search with gaze-contingent masks in 3D, omnidirectional virtual every-day scenes. As such, our analyses of visuo-motor measures inform about search behaviors in more natural settings, in particular about the role of eye and head movements.

Overall, we have found that the impact of gaze-contingent windows and masks on visual search in VR were quite similar to those reported previously in the literature using on-screen experiments. Eye rotation movements increased in amplitude when we removed central vision and decreased without peripheral information, as was expected from previous findings [8,20,21,22,23,24,47,53,54]. This effect translated to gaze movements as well, though the increase in eye rotation amplitude was mediated by the baseline increase observed without masking. During no-mask trials observers made longer saccades than observed in free-viewing in VR [47], the impetus to accomplish the task pushed them to explore their omnidirectional environment with more distance between fixations. The absence of central information increased backward saccade rates as observers made back-and-forth movements between regions of interest [21,24,47,55]. Either in an effort to verify a target exra-foveally or because the lack of central information leads visual attention to be grasped in a bottom-up fashion by peripheral stimulations. On the contrary, with no peripheral stimulation gaze was guided predominantly by scene exploration, showing a strong saccadic momentum [78,79]. We observed that fixation durations were only slightly affected by masking, significantly so when peripheral vision was missing. Contrary to gaze-contingent experiments manipulating spatial frequencies (e.g., [8,20,21,22,23,52,53,54]) or color [17], our protocol removed all stimulation within masked areas. As a result, there was no difficulty related to processing modified information since it was entirely removed. There may exist difficulties in planning new saccades in the peripheral masking condition but this did not transpire in this study. As was observed previously [24,47], participants made shorter fixations of longer amplitudes generally directed in a common direction, characteristic of a global scanning phase [14,15].

In addition to merely analyzing eye rotations in 3D, VR gave us the opportunity to study how the head contributes to visual search in 3D. Head movements serve exploration by extending saccade amplitude’s length, being fairly simple and moreover showing a very strong forward momentum bias. Note that this bias also appeared because we defined saccades on the basis of gaze movements, in a case where the head rotates to span a big part of one’s surroundings there may be several saccades happening. Because we segmented all signals into fixations on the basis of the overall gaze movement data, a large head movement would be segmented into several ones corresponding to the motion of the head in-between two fixations. Therefore, one head movement made to explore a big part of the scene will be recorded as several head movements in the same direction (forward relative directions). Additionally, head movement is measured by the VR headset’s sensors not differentiating rotations stemming from feet, torso or neck movements. The effect of transient vision loss on head movements is new and shown as a global decrease in average amplitude, we assume that this effect reflects the abnormality of the situation. If we consider that effects of vision loss on eye movements is stimulus-driven and almost uncontrollable, reducing head movements may reflect difficulty through uncertainty and lack of proper control of the situation.

We have reported differences between visual search and free-viewing tasks by comparing our results to previous research on-screen and in VR. In particular, we show longer saccade amplitudes in this study compared to free-viewing in VR; possibly reflecting a bias toward exploration, which is needed to scan a large environment for a target with a time-constraint. This partly replicates the findings of Mills et al. [93] comparing four different visual tasks. The authors measured shorter fixations and longer saccades in the visual search task compared to free-viewing. Smith and Mital [94] reported the same effects comparing free-viewing to a “spot-the-view” task; free-viewing resulted in longer fixations and shorter saccades. Considering the literature, it makes sense to discern free-viewing from goal-oriented tasks. Importantly, our results do not show shorter fixation durations compared to a previous VR free-viewing experiment [47], instead we possibly showed a tendency of immersive contents to decrease average fixation durations. A dedicated study comparing visual search and free-viewing is needed to make clear interpretations with regard to task-related differences of eye movements in VR.

Studying the time-course of fixation duration and movement amplitudes during saccades we showed that participants started trials with the head hardly moving. We believe this mirrors the initiation phase of visual search: the time needed to start significantly shifting the field of view to explore may be positively correlated with the difficulty to control the situation and build a suitable representation of the scene. Right after the first fixation on a target (start of the verification phase), observers made shorter head movements because there was no need to make big visual field shifts once a potential target was in view. Our analyses confirm our hypothesis by showing that scanning phases are exploratory with long saccades and short fixation durations, while verification phases are characterized by short saccades and long fixation durations. Fixation durations particularly increased at the end of the second phase as participants made their final decision before pressing the controller’s button to end the trial. Our data did not show a clear interaction effect between search phases and masks. Interestingly, eye rotation amplitudes were not significantly impacted by search phases. We noticed that scanning phases began with eye movement amplitudes larger than observed during the rest of the phase, which had been observed on-screen previously [95,96] and is thought to reflect a strategy to extract the most informative part of the scene at the start of a trial. This strategy turns into exploration of the omnidirectional scene when head movements increase in amplitude as eye movements become smaller. Considering return saccade rates (SRR), we observed that central masking only affected this measure during verification phases. This indicates that scene exploration is not hindered by a large central mask.

The strong effect of transient peripheral vision loss on almost all measures studied reiterates the importance of peripheral vision in visual search—particularly in real-world scenes—to build a representation of the scene and to detect potential targets. Masking central vision did not have as strong an impact on visuo-motor behaviors as we would expect from past literature considering the large mask size. This implies that peripheral vision (6 deg. and up) is in some measure sufficient to explore a complex environment.

The differential effects of vision loss on eye and head movements potentially indicate a difference in control and planning during fixations and saccades. Eye movements appear to be strongly impacted by available visual information, while head rotations are dedicated to exploring the environment. In peripheral-masking trials the head even allows planning saccades into the masked area. This shows that there can be a strong effect of vision loss and at the same time a pursuit of a task goal (exploration), both dependent on different elements of the visuo-motor system. Further work on the topic of visual search in virtual reality could be beneficial to the quality of life of patients suffering from visual field deficits [97,98,99] for example, by recommending efficient search strategies based on the remaining useful field of view.

## 5. Conclusions

In this study, thanks to a fully integrated VR eye tracking apparatus, we were able to gain new insights regarding gaze behaviors during an object search task in quasi-natural settings. We replicated on-screen gaze-contingent experiments and made clear that the impact of gaze-contingent masking is strongly reflected on eye movements, less so on head rotations. We have shown that the head, not the eyes, drives exploration and analyses during the scanning and verification phases, respectively. We hope this study will motivate the community to look into VR to simulate more natural conditions and implement new experimental designs.

## Figures and Tables

**Figure 1 brainsci-10-00841-f001:**
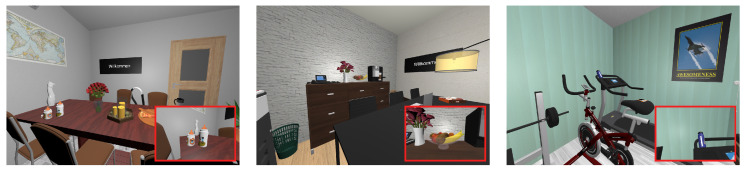
Views from three virtual rooms. Examples of target items (from left to right: baby bottles, fruit bowl, drinking bottle) appear within red frames. The black display screen appears in the first two rooms as an illustration of its position only (it is not actually visible in rooms).

**Figure 2 brainsci-10-00841-f002:**
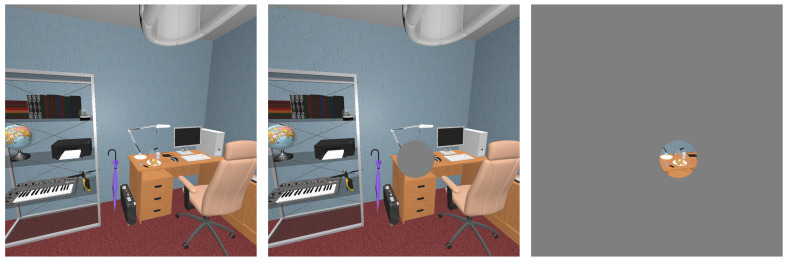
Masking conditions are presented here in a viewport measuring 90 by 90 deg. of field of view, mask radii are proportionally accurate. From left to right: control no-mask condition, central mask of six deg. of radius, peripheral mask of six deg. of radius.

**Figure 3 brainsci-10-00841-f003:**
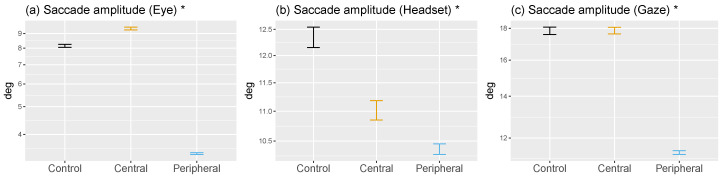
Visuo-motor variables are presented in this figure as a function of mask condition (control in black, central in orange and peripheral in blue) showing means and 95%CI. An asterisk to the right of a variable’s name indicates that it was log-transformed in linear mixed models and is presented on a log-scale here. SRR stands for saccadic reversal rate and measures the ratio of backward saccades produced precisely in the 170–190 deg. interval.

**Figure 4 brainsci-10-00841-f004:**
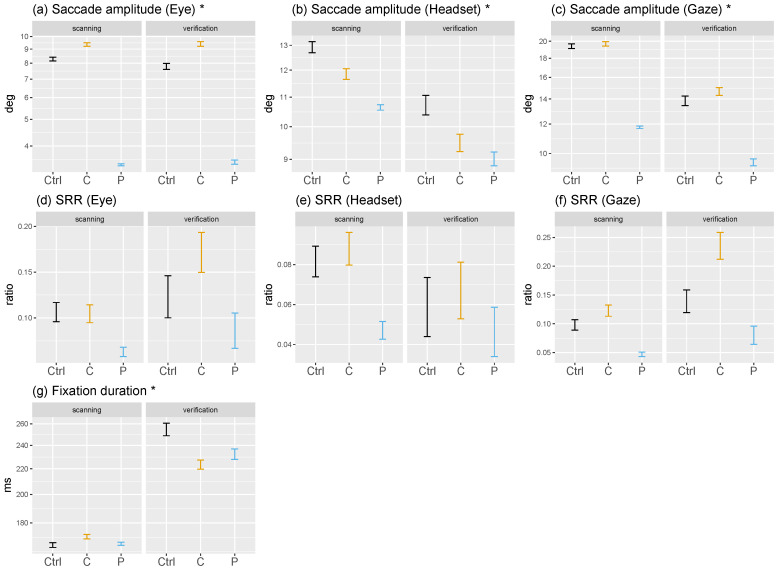
Visuo-motor variables are presented in this figure as a function of search phase (scanning and verification) and mask condition (control in black, central in orange and peripheral in blue) showing means and 95%CI. An asterisk to the right of a variable’s name indicates that it was log-transformed in linear mixed models and is presented on a log-scale here. SRR stands for saccadic reversal rate and measures the ratio of backward saccades produced precisely in the 170–190 deg. interval.

**Figure 5 brainsci-10-00841-f005:**
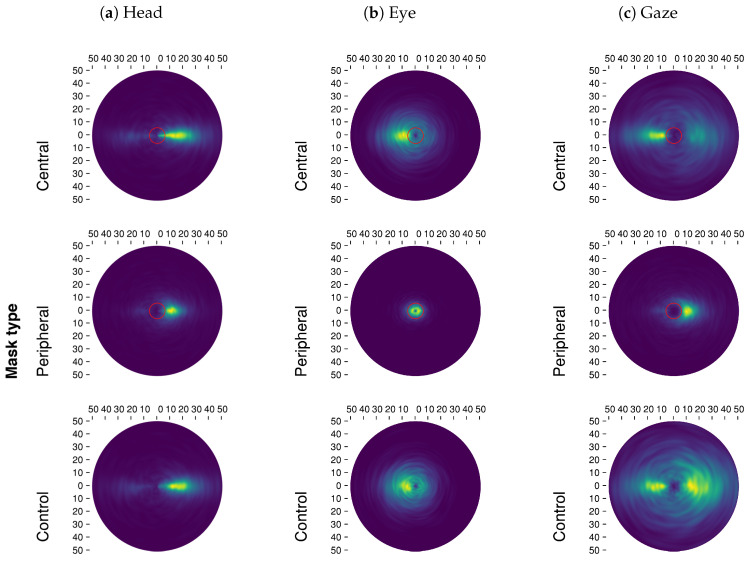
The joint distribution of saccade amplitudes and relative saccade directions are presented in this figure for eye, head and gaze as a function of mask conditions. Red circles at the center of the polar plots denote masks’ radius (six deg.). A brighter color represents a higher density of saccades.

**Figure 6 brainsci-10-00841-f006:**
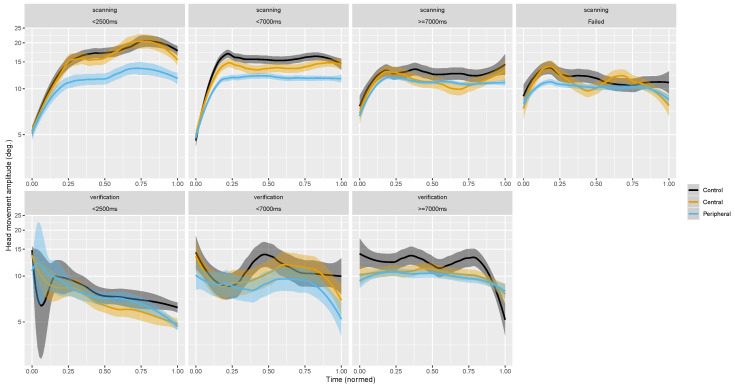
Time-course of average head movements amplitude during saccades as a function of search phase (trial duration subgroups do not overlap, see Figure A3, Appendix D). Ribbons show 95%CI. Control condition appears in black, central masking in orange and peripheral masking in blue.

**Table 1 brainsci-10-00841-t001:** Statistical output of the analysis of the effects of search phases on fixation duration and saccade amplitude (hypotheses *Hb*: scanning vs. verification phase).

Variable	Masking	*b*	SE	*t*
Saccade amplitude (Eye)	Control	−0.06	0.01	−4.05
	Central	−0.01	0.01	−1
	Peripheral	0.04	0.01	3.43
Saccade amplitude (Head)	Control	−0.2	0.02	−12.14
	Central	−0.21	0.01	−14.96
	Peripheral	−0.16	0.01	−12.25
Saccade amplitude (Gaze)	Control	−0.35	0.01	−24.04
	Central	−0.3	0.01	−23.69
	Peripheral	−0.22	0.01	−18.53
SRR (Eye)	Control	0.02	0.01	1.61
	Central	0.06	0.01	6.56
	Peripheral	0.02	0.01	2.2
SRR (Head)	Control	−0.02	0.01	−2.99
	Central	−0.02	0.01	−2.85
	Peripheral	0	0.01	0.19
SRR (Gaze)	Control	0.04	0.01	4.22
	Central	0.11	0.01	11.8
	Peripheral	0.03	0.01	3.37
Fixation duration	Control	0.44	0.01	42.51
	Central	0.27	0.01	30.04
	Peripheral	0.35	0.01	42.01

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
