# Peer review of "Effects of Transient Loss of Vision on Head and Eye Movements during Visual Search in a Virtual Environment"

_brainsci, 2020, doi:10.3390/brainsci10110841_

Round 1

Reviewer 1 Report

The authors used a gaze-contingent protocol (window/scotoma) to investigate the contribution of central and peripheral vision in a visual search task involving 3D scenes. They use a virtual environment with a head mounted device allowing the participant to move freely in scenes chosen to fit in the physical room where the experiment takes place. Eye and head movements are recorded during two phases of the visual search task: exploration to find the target and verification that the object corresponds to the target. The results show that the impact of gaze-contingent windows and scotomas on visual search in virtual environments are similar to those reported on screens in the literature: The exploration phase involves long saccades and short fixation durations, while verification phases are characterized by shorter saccades and longer fixation durations. Peripheral vision loss is more detrimental than central vision loss. Head movements drive exploration.

The study is interesting, well written, and presents two main novelties compared to visual search tasks on screen: the use of 3D scenes and a measure of head movements. The result section is detailed and interpreted  adequately.

The authors conclude that their findings “provide more information concerning how

knowledge gathered on-screen may transfer to more natural conditions, and attest to the experimental usefulness of eye tracking in virtual reality.” Yet, it seems that most of the results in terms of eye movements are similar in classical paradigms on screens and in VR. Except for the technical masking aspect what is the added value of using VR to study eye movements in visual search compared to head mounted devices (e.g., glasses) in natural scenes (e.g., Sippel et al 2014 in a supermarket)?

Would  VR  help the rehabilitation of patients with central or peripheral vision loss?

-Why use a word rather than a photo of the target?

Some information is missing in the method:

- Did you control the semantic congruity of target objects in the scene?

- Was the target always present?

- Did the target appear equally in the central and in the peripheral part of the scene?

Author Response

Please see the attachment for the author's reply to the reviewer.

Please follow the link below to obtain an updated version of the manuscript with modifications highlighted in green.

https://hessenbox-a10.rz.uni-frankfurt.de/getlink/fi2kHrBRMFu3g3r7qeUbk6gs/DavidBeitnerMLVO_20_modif.pdf

Reviewer 2 Report

The manuscript starts with the assumption that eye-movements are made to process information (Line 21), hence to form perception in my understanding. However, it then came to the conclusion that "eye movements are driven by perception" (Line 13 & 420). This seems confusing and circular to me. Considering the statement “eye movements are driven by perception” was made in contrast to “head movements serve a higher behavioral goal”, I assume that it meant the “stimulus-driven vs. goal-directed” visual processing. So a different wording such as "physical stimuli" or "visual stimulation" might be more appropriate to replace "perception" here.

In the introduction, the three temporal phases during visual search were conceptually defined based on 2D eye-tracking data (i.e., head movements are not involved). However, I don’t see how they were technically defined and obtained in the present study. Were they determined by gaze data or by all three kinds of data (eye, head, and gaze)?

Line 278-279, the two search phases were sorted into durations of < 2.5s, < 7s, and > 7s, but I don’t see a priori for the different time ranges. Also, does “less than 7 sec.” mean “less than 7 secs while over 2.5 secs”?

I find it hard to refer the figures, especially the figures concerning the results, to the descriptions in the main text. It would be nice if the authors can cite all the figures and the table in the main text.

This study addressed an interesting issue with up-to-date VR technologies. The strength of this study lies in that the results contribute to teasing apart the role of head movements and eye movements in both free viewing and restricted viewing (simulated central and peripheral vision loss). The limitation of this study is that the results are somewhat descriptive and the interpretations hence are somewhat speculative. Nevertheless, I am supportive of it being published after a minor revision, as the results add to the few VR studies in the field.

Author Response

(The authors gave the same response as above.)
